# *Allium hookeri* Root Extract Inhibits Adipogenesis by Promoting Lipolysis in High Fat Diet-Induced Obese Mice

**DOI:** 10.3390/nu11102262

**Published:** 2019-09-20

**Authors:** Hyun Ju Kim, Min-Jung Lee, Ja-Young Jang, Sung-Hyen Lee

**Affiliations:** 1Research and Development Division, World Institute of Kimchi, Nam-Gu, Gwangju 61755, Korea; 2Functional Food & Nutrition Division, Department of Agro-food resources, Rural Development Administration, Wanju, Jeonbuk 55365, Korea

**Keywords:** *Allium hookeri* root, anti-obesity, adipogenesis, lipolysis, high fat diet

## Abstract

*Allium hookeri* (AH) is widely consumed as a herbal medicine. It possesses biological activity against metabolic diseases. The objective of this study was to investigate effects of AH root water extract (AHR) on adipogenesis in 3T3-L1 cells and in high-fat diet (HFD)-induced obese mice. AHR inhibited lipid accumulation during adipocyte differentiation by downregulation of gene expression, such as hormone sensitive lipase (HSL), lipoprotein lipase (LPL) and an adipogenic gene, CCAAT/enhancer binding protein-α in 3T3-L1 preadipocytes. Oral administration of AHR significantly suppressed body weight gain, adipose tissue weight, serum leptin levels, and adipocyte cell size in HFD-induced obese mice. Moreover, AHR significantly decreased hepatic mRNA expression levels of cholesterol synthesis genes, such as 3-hydroxy-3-methylglutaryl CoA reductase, sterol regulatory element-binding transcription factor (SREBP)-2, and low-density lipoprotein receptor, as well as fatty acid synthesis genes, such as SREBP-1c and fatty acid synthase. Serum triglyceride levels were also lowered by AHR, likely as a result of the upregulating gene involved in fatty acid β-oxidation, carnitine palmitoyltransferase 1a, in the liver. AHR treatment activated gene expression of peroxisome proliferator-activated receptor-γ, which might have promoted HSL and LPL-medicated lipolysis, thereby reducing white adipose tissue weight. In conclusion, AHR treatment can improve metabolic alterations induced by HFD in mice by modifying expression levels of genes involved in adipogenesis, lipogenesis, and lipolysis in the white adipose tissue and liver.

## 1. Introduction

Obesity is associated with insulin resistance and dyslipidemia, which can lead to metabolic diseases, and is a concern that is growing in prevalence worldwide [1]. The main feature of obesity is hypertrophy of white adipose tissue (WAT) caused by energy imbalance [2]. An imbalance between energy intake and expenditure results in adipose tissue enlargement due to excessive lipogenesis and scanty lipolysis [3]. WAT is a complex organ that secretes various endocrine factors such as adiponectin, estrogen, leptin, and an array of cytokines. It plays a central role in energy homeostasis control [4]. Excessive fat accumulation in the adipose tissue results in obesity-associated metabolic complications [3]. Some principal mechanisms of lipid metabolism are modulated by insulin signaling, adipogenesis, adipocyte differentiation, lipolysis, and β-oxidation of free fatty acid (FFA) in the obese state. Lipolysis refers to the degradation of triglyceride (TG) to yield FFA and glycerol. It can be induced by lipase, such as adipose triglyceride lipase (ATGL) and hormone-sensitive lipase (HSL) in adipocytes [5,6], which are the direct transcriptional target of peroxisome proliferator-activated receptor (PPAR)-γ [7]. Excess FFAs produced by lipolysis can migrate in the circulation and accumulate in the blood or the liver, resulting in upregulation of proteins involved in de novo lipogenesis, such as sterol regulatory element-binding protein (SREBP1) 1 and fatty acid synthase (FAS) [8]. Conversely, lipid accumulation can be reduced by upregulation of genes involved in β-oxidation of FFA, such as PPAR-α and mitochondrial carnitine palmitoyltransferase (CPT)1, in the adipose tissue and liver [9]. For this reason, controlling these factors has been an ideal therapeutic strategy to prevent or ameliorate obesity associated complications. Furthermore, the use of natural products as therapeutic agents in treating and preventing obesity has become popular [10]. 

*Allium hookeri* Thwaites (*Liliaceae* family, AH) has been widely cultivated for food or medicinal usages in South Asia and Southeast Asia [11]. AH leaf or root has been used as food or food ingredients such as kimchi, yogurt, seasonings, and even instead of onion and garlic in Korea. AH contains plenty of nutrients and has especially high levels of sulfur-containing compounds such as allicin, *S*-allylcysteine, and cycloalliin [12]. Beneficial effects of AH are attributed to a variety of phenols (ferulic acid, gallic acid and cinnamic acid), phytosterols, linoleic acid and organosulfur compounds [13]. AH is known to have biological properties, such as being antioxidant [14,15,16], antimicrobial [17], antidiabetic [18,19,20], anti-obesity [21,22], and anti-inflammatory [23,24,25], as well as hepatoprotective effects [26]. It can also improve bone formation [27], gastritis [28], and immune responses [29]. Despite the many biological activities of AHR, its unique flavor as a result of sulfur compounds, makes it less appetizing as a form of food. Therefore, Yang et al. [30] recommended that hot-air dried AHR was a better solution in terms of flavor and, consequently, the anti-obesity effect. Our previous study demonstrated that AHR exhibits anti-inflammatory through NF-*κ*B down-regulation in LPS-induced macrophages [25] and anti-diabetic activity through the inhibition of pancreatic injury in type 1 diabetic rats [20]. However, mechanistic study of AHR on adipogenesis and lipolysis in adipocyte and white adipose tissue of high fat diet (HFD)-induced mice remains unresearched. Therefore, the objective of this study was to investigate the effect of AHR on adipogenesis in 3T3-L1 adipocytes and expression levels of genes involved in lipogenesis and lipolysis in HFD-induced mice in order to elucidate the underlying mechanism. Our data revealed that AHR induced lipolysis via upregulation of HSL and LPL, although diglyceride acyltransferase (DGAT) and PPAR-γ were upregulated in WAT. AHR also suppressed lipogenesis via downregulation of SREBP-1c, 3-hydroxy-3-methylglutaryl (HMG)-CoA reductase, and low-density lipoprotein (LDL)-receptor in the liver. Conversely, hepatic lipid accumulation is reduced by upregulation of genes, such as CPT1a, involved in β-oxidation of FFA. 

## 2. Materials and Methods

### 2.1. Sample Preparation and Regents

AH was cultivated in Sunchang-gun, Jeollabuk-do, Korea. Plant identification was done in the Sunchang Agricultural Development and Technology Center. AH (RDAAH15) was deposited at Rural Development Administration. Freeze-dried (PVTFD 10R, Ilsin Lab, Yangju, Korea) AHR was suspended with ten volumes of distilled water and extracted for 12 h in a water bath at 95 °C. Residues were reextracted twice under the same conditions. These hot-water extracts were filtered through Whatman filter paper (grade No. 2; Whatman International, Kent, UK), freeze-dried, pulverized, and stored at −75 °C until further analysis. 

Cell culture reagents including Dulbecco’s modified Eagle’s medium (DMEM), bovine calf serum (BCS), fetal bovine serum (FBS), penicillin–streptomycin (P/S), insulin, and trypsin/EDTA were purchased from Gibco (Gaithersburg, MD, USA). Dexamethasone, 3-isobutyl-1-methylxanthine (IBMX), isopropanol, Oil Red O, and chemical reagents were mainly purchased from Sigma-Aldrich (St. Louis, MO, USA). 

### 2.2. Cell Culture and Differentiation

Mouse 3T3-L1 preadipocytes were purchased from Korean Cell Line Bank (Seoul, Korea). In brief, cells were cultured in DMEM supplemented with 10% bovine calf serum at 37 °C in a humidified atmosphere of 5% CO_2_ for one day. After cells reaching confluence (designated “day 0”), cell differentiation was induced with a mixture (DMI) of 0.5 mM 3-isobutyl-1-methylxanthine (MIX), 100 µM indomethacin, 0.25 µM dexamethasone (DEX), and 167 nM insulin in DMEM containing 10% FBS. MIX, DEX, and indomethacin were obtained from Sigma-Aldrich (St. Louis, MO, USA). The medium was changed every two days. Cells were treated with 0, 100, or 500 µg/mL of AHR extract for the last 3 days of the differentiation period. To analyze cytotoxicity of AHR, cell viability was evaluated using 3-(4,5-demethylthiazol-2-yl)-2, 5-diphenyltetrazolium bromide. 

### 2.3. MTT Assay

Cytotoxicity of 3T3-L1 preadipocytes was measured by CCK-8 kit. The 3T3-L1 preadipocytes were placed in 1 × 10^4^ cells/well and incubated with N or AHR (0, 100, 500 µg /mL) for 2 days. Thereafter, cells were washed with phosphate-buffered saline (PBS) three times. Then, cells were reacted with 10 µL of CCK-8 solution for 2 h. Absorbance was read at 450 nm.

### 2.4. Real-Time Polymerase Chain Reaction (RT-PCR) Analysis

RNA was isolated from 3T3-L1 adipocytes using an RNeasy plus Mini Kit (Qiagen, Valencia, CA, USA) according to the manufacturer’s protocol. The extracted RNA was reverse transcribed into cDNA using a high-capacity cDNA reverse transcription kit (Applied Biosystems, Foster City, CA, USA). Then RNA expression level was quantified by RT-PCR using SYBR Green PCR Master Mix (Applied Biosystems, Woolston, Warrington, UK) and the 7500 Real Time PCR system (Applied Biosystems, Foster City, CA, USA) according to the manufacturer’s protocol. RT-PCR was performed using gene-specific primers (Table 1). Amplification cycles were: denaturation at 95 °C for 50 s, annealing at 55 °C for 1 min, and elongation at 72 °C for 50 s for each gene. Gene expression was normalized against the expression level of β-actin. Relative gene expression level was calculated using the 2^−ΔΔCt^ method. 

### 2.5. Oil Red O Staining and Triglyceride Assay

Cells were washed with cold phosphate buffered saline (PBS) and fixed with 4% (*v*/*v*) formalin for 30 min on Day 8. After Oil Red O staining, cells were photographed using a phase-contrast microscope (DMIR; Leica Microsystems, Wetzlar, Germany) at 100× magnification. During adipocyte differentiation (Days 7–8), cells were treated with AHR at concentrations of 0, 100, and 500 µg/mL in 6-well plates for 2 days. On Day 9, to analyze contents of cellular triglycerides, cells were washed with PBS, scraped into 200 µL PBS, and homogenized by sonication for 1 min. Cell lysates were assayed for triglycerides, using assay kits (Sigma-Aldrich, St. Louis, MO, USA), according to the manufacturer’s instructions.

### 2.6. Animal Studies

Six-week old male C57BL/6J mice were purchased from Jung Ang Lab Animal Inc (Seoul, Korea). After one week of acclimation, mice were randomly divided into four groups: (1) normal diet (ND, 15% of fat calories), (2) high-fat diet (HFD, 45% of fat calories), (3) another HFD group that also received AHR 100 mg/kg B.W./day, and (4) 500 mg/kg B.W./day in drinking water once a day, respectively. During the experiment, water intake and body weight were recorded weekly to adjust the dose of AHR. Each group consisted of 6–8 mice. All animals were housed at a room temperature of 20 ± 2 °C and humidity of 50 ± 5% with a 12 h light/dark cycle. They were given free access to food and water during the entire experimental period. These animals were maintained on a feeding program for nine weeks. Food intake and body weight were measured weekly using a digital weighing scale. All animal studies were approved by the Institutional Animal Care and Use Committee (IACUC) of the World Institute of Kimchi (Approval No. WIKIM-IACUC 201601).

### 2.7. Serum Biochemical Analyses

Serum triglyceride (TG) and total cholesterol (TC) levels were measured using commercial enzyme kits (Asan Pharmaceutical Co., Seoul, Korea). Serum leptin and adiponectin levels were measured with a leptin ELISA kit (Enzo, Newyork, NY, USA) and an adiponectin ELISA kit (Abcam, Inc., Cambridge, UK), respectively. A microplate reader (Infinite F200, Tecan Japan Co., Ltd., Kanagawa, Japan) was used for absorbance measurement.

### 2.8. Gene Expression Analysis

To extract RNAs from tissues, 20 mg of liver was used and extracted by RNAiso PLUS (Takara Bio, Otsu, Japan). RNA concentration was measured using a nanodrop. First strand cDNA was synthesized using an AMPIGENE cDNA synthesis Kit (Enzo, Farmingdale, NY, USA). cDNA was synthesized with at 42 °C for 30 min followed by a treatment at 85 °C for 10 min to deactivate the reverse transcriptase. Real-time PCR was performed by mixing 10 µL TOPreal qPCR 2 × PreMIX (Enzynomics, Daejeon, Korea) with 1 µL cDNA, 2 µL primer, and 7 µL nuclease free water. Nucleotide sequences of the primers used and the PCR conditions are shown in Table 2.

Total RNA was isolated from WAT and diluted to be at a concentration of 100 ng/5µL. Frozen reporter CodeSet and probeset were dissolved on ice, inverted several times, and spun down. Then 70 µL of hybridization buffer was added into the reporter CodeSet tube to make a master mix. Capture probes were plunged into each well (8 µL/well). Then 5 µL of RNA was added to each well. Before starting the reaction, the thermocycler was pre-heated to 65 °C with a reaction volume of 15 µL. The nCounter prep-station was loaded with a sample that was hybridized, washed, and immobilized. After the above procedure was completed in the prep-station, the cartridge was counted on a digital analyzer and the degree of expression was analyzed using nSolver software. Information on target genes is shown in Table 3.

### 2.9. Histological Analysis

Slices of epididymal adipose tissue was immediately fixed in 10% (*v*/*v*) neutral formalin and embedded in paraffin wax. Sections (4–6 mm thick) were cut. Each section was then stained with hematoxylin and eosin (H & E) and observed with a microscope (BZ-9000, Keyence, Osaka, Japan). Each section was used as an index to observe the morphology of adipocytes.

### 2.10. Statistical Analysis

All data are expressed as mean ± standard error of the mean (SEM). Comparisons between experimental groups were made using one-way analysis of variance (ANOVA), following Duncan’s multiple range-test (SPSS 12.0 software, Chicago, IL, USA). Each value was the mean of at least three separate experiments for each group. Mean values were considered significantly different when *p* values were less than 0.05.

## 3. Results

### 3.1. Fat Accumulation and mRNA Expression in 3T3-L1 Adipocytes

To investigate whether AHR reduced adipocyte differentiation, we first assessed AHR for its toxic effect using MTT assay. As shown in Figure 1A, AHR was not cytotoxic to 3T3-L1 adipocytes at a concentration range of 0–500 µg/mL. We therefore decided to use AHR at concentration of 100 or 500 μg/mL during adipocyte differentiation. To assess the inhibitory effect of AHR on adipogenesis, we differentiated 3T3-L1 preadipocytes into adipocytes with a differentiation cocktail (MDI) in the presence or absence of AHR for 8 days. As shown in Figure 1B,C, AHR significantly reduced fat accumulation and intracellular TG contents based on Oil Red O staining. AHR also suppressed gene expression involved in adipogenesis (C/EBP-HSL and LPL by RT-PCR analyses (Figure 1D). No statistical differences were found in PPAR-γ, ATGL and SREBP-l between non-treated and AHR treated adipocytes. These data suggested that the reduction of fat accumulation by AHR was related to the decreased expression of genes involved in adipogenesis.

### 3.2. Body and Tissue Weights

To confirm the anti-obesity effect of AHR, body weights during the experiment were measured. HFD-fed mice gained more body weights than AHR-treated mice (Figure 2A). However, there was no statistical significance in body weight gain or food intake between AHR-treated group and HFD-fed group. Also, the food efficiency ratio of the AHR-treated group was lower than that of the HFD-fed group (Table 4). As shown in Figure 2B, AHR-treated mice gained substantially less abdominal and epididymal white adipose tissue than HFD-fed mice. Liver weights of HFD-fed mice were significantly higher than those of normal mice, while those were again lower in AHR-treated mice. However, other tissue weights were not affected by the HFD or AHR treatment (Table 5).

### 3.3. Cell Size of Epididymal Adipose Tissue

Due to the differences in adipose tissue weights, tissue sections were stained with H&E and adipocyte sizes were observed. HFD induced severe hypertrophy of epididymal adipocytes which showed a significant increase of 58%, as compared to the normal group. On the other hand, AHR100 and AHR500 groups showed significant decreases in cell size of epididymal adipocytes, by 21%, and 34%, respectively, as compared to the HFD group, indicating that adipose tissue hypertrophy due to the feeding of HFD could be suppressed by administration of AHR (Figure 3).

### 3.4. Serum Measurements

Disorders in lipid metabolism are commonly observed in patients who are obese, including elevated serum triglyceride and cholesterol levels [2]. To determine whether AHR treatment could ameliorate obesity-related abnormalities in blood parameters, we analyzed circulating total cholesterol, triglyceride, leptin, and adiponectin levels. Total cholesterol and triglyceride concentrations in plasma were lower in the AHR500 group than those of the HFD-fed group, although total cholesterol levels were not significantly lower in the AHR500 group (Table 6). AHR treated mice had lower serum leptin levels, but higher serum adiponectin levels than the HFD-fed mice (Figure 4). Leptin is an adipose tissue hormone. Increases of plasma leptin levels can lead to reduction of food intake and loss of weight [31]. Moreover, while a defect in leptin signaling is associated with hyperphagia and a marked decrease in energy expenditure in mice, it mainly increases appetite in humans, resulting in obesity [31].

### 3.5. mRNA Expression Level in the Liver

To investigate the mechanism by which lipogenesis in the liver was reduced by AHR treatment, hepatic expression levels of genes associated with lipid metabolism were measured by RT-PCR. Regarding genes associated with cholesterol synthesis, the mRNA levels of HMG-CoA reductase, SREBP-2, and LDL receptor were decreased by AHR treatment (Figure 5). Expression levels of lipogenic genes such as SREBP-1c and FAS were also decreased after AHR treatment, as compared to those in the control HFD-fed mice. Thus, AHR promoted oxidation of FFA by upregulating CPT-1a expression in the liver, thus reducing the circulating concentrations of TG in HFD-fed mice (Figure 6).

### 3.6. mRNA Expression Level in Epididymal Adipose Tissue

To investigate the mechanism by which AHR reduced fat deposit size in epididymal adipose tissue, expression levels of lipid metabolism-associated genes were determined by nCounter gene expression assay (Table 7). When mRNA levels associated with TG lipolysis were measured, HSL and LPL mRNA levels were found to be significantly increased after AHR500 treatment. DGAT mRNA level was significantly higher in the AHR500 group than that of the HFD group. PPAR-γ known to be involved in adipocyte differentiation and miniaturization was also increased by AHR treatment. Although mRNA levels of perilipin and SCD1 related to adipocyte lipolysis were slightly increased after AHR500 treatment, such increases were not statistically significant. AHR treatment did not affect PPAR-*γ* or CPT1a gene expression in adipose tissue. Other genes in epididymal adipose tissue of HFD-induced mice were not affected by AHR treatment. These findings clearly demonstrated that AHR treatment could augment lipolysis and miniaturization of adipocyte in diet-induced obesity condition.

## 4. Discussion

AH is used as a traditional medicinal plant that is widely distributed in Southeast Asia [11]. AH has been widely cultivated in South Korea since 2012. It has a variety of phenols such as protocatechuic acid, ferulic acid, gallic acid, and syringic acid and especially has high levels of sulfur-containing compounds such as alliin and cycloalliin compared to other *Allium* species [12,26]. AH has received considerable attention owing to its well-documented physiological activities [14,15,16,17,18,19,20,21,22,23,24,25,26,27,28,29,32]. Our previous study has demonstrated that major components of AHR are alliin, *S*-allylcysteine, and phenolic compounds that are known to have anti-inflammatory [23,24,25] and antidiabetic effects [18,19,20]. A metabolomic study has shown that 25 metabolites altered by HFD-induced hyperlipidemia can be restored by AHR administration [33]. However, little is known about the impact of AHR on underlying cellular mechanism of adipogenesis, lipogenesis and lipolysis in HFD-induced obesity up to date. We found that AHR partially inhibited adipogenesis in 3T3-L1 adipocyte and promoted lipolysis in WAT of HFD-fed mice.

In this study, AHR treatment inhibited lipid accumulation, which was associated with downregulation in mRNA expression of C/EBPα, LPL, and HSL in 3T3-L1 adipocytes. Differentiation of adipose tissue involves upregulation of key adipocyte-specific transcription factors, such as C/EBPα and PPAR-γ [34], that can promote energy storage in the form of TG. ATGL and HSL hydrolyzed TG to FFA and glycerol release in adipose tissue and key factor for the fat mobilization of adipose tissue [35]. LPL is secreted by mature adipocyte and the rate-limiting enzyme for the import of TG. It plays a central role in controlling lipid accumulation. Decreased HSL may show marked accumulation of diacylglycerol and absence of glycerol release and FFA from adipose cells [36]. These results suggest that the decrease in the generation of FFA might prevent differentiation of adipose cells. Phenolic acid, such as gallic acid, suppressed lipid accumulation in 3T3-L1 adipocytes by downregulating the mRNA expression of C/EBPα, PPAR-γ, and FAS, as well as stimulating the mRNA expression of perillipin, fatty acid binding protein (FABP)-4, and glucose transporter type (GLUT)-4 [37]. TG stored in visceral fat can be hydrolyzed by lipolytic enzymes such as LPL and HSL to liberate glycerol and FFA [38]. We found that AHR could promote mRNA expression of these enzymes by activation of PPAR-γ in WAT of HFD-induced mice, which could reduce adipocyte size. FFA is derived from lipolysis in adipocyte and it is released into circulation and transported to the liver or muscle, which can be utilized for β-oxidation or lipogenesis [39,40]. However, uptake of excessive amounts of FFA by alteration of lipolysis in these tissues causes obesity and insulin resistance [41,42]. Regarding lipid metabolism, some studies have shown that AH has pronounced effects on obesity by down-regulating GLUT-4 in adipocytes and hepatic mRNA levels of FAS and LPL in diet-induced obese mice [21,22]. In this study, AHR decreased mRNA expression involved in hepatic lipogenesis and increased mRNA expression involved in adipogenesis and lipolysis in WAT, in spite of a reduction in weight and adipocyte size of WAT. Our results revealed that AHR-treated mice showed lower expression levels of hepatic cholesterol synthesis genes such as HMG-CoA reductase, SREBP-2, and LDL-receptor than HFD-fed control mice. In addition, AHR intake decreased the expression of de novo lipogenesis medicated by SREBP-1 and FAS and increased the expression of CPT1, resulting in FFA β-oxidation.

The central finding of the present study was that AHR suppressed the differentiation of 3T3-L1 adipocytes and reduced adipocyte size in white adipose tissue of HFD-induced mice. To elucidate the underlying mechanism these effects, expression levels of genes associated with adipogenesis, lipogenesis and lipolysis in the white adipose tissue were measured. Results showed that AHR-induced mRNA expression of PPAR-γ in adipose tissue promoted lipolysis mediated by HSL and LPL. PPAR-γ is a transcription regulatory factor that regulates adipocyte proliferation, lipogenesis, and lipolysis [43]. PPAR-γ agonists can reduce the number of hypertrophic adipocytes and increase the number of miniaturized adipocyte cells in rats fed HFD [44]. Therefore, PPAR-γ is considered to be an orchestrator in adipogenesis. It plays an important role in suppressing excessive adipocyte hypertrophy and promoting normalization of adipocyte size [44]. HSL, a major enzyme for hydrolysis of TG, is up-regulated by PPAR-γ. A lack of HSL leads to insulin resistance and obesity [45]. On the other hand, reductions of PPAR-γ decreased adipocyte size and WAT mass by activation of FFA-oxidation and energy expenditures, thereby alleviating insulin resistance and obesity [46]. Generally, it is well accepted that adipocyte hypertrophy in an obese state is associated with macrophage infiltration and elevated inflammatory cytokine secretion [47,48]. Therefore, the suppression of adipocyte hypertrophy is important for preventing obesity and its complications. It has been shown that AHR can alleviate oxidative stress-induced inflammatory responses [25], insulin depletion, and β-cell damage in pancreas of diabetic animals [18,19,20]. To describe the metabolic fate of FFA, we measured mRNA expression levels of CPT1β, FABP4 and PPAR-α known to be responsible for the influx of FFA into mitochondria and advancement of β-oxidation. We also measured mRNA expression of DGAT-1 known to catalyze the final step of TG synthesis. In our study, AHR did not significantly affect the mRNA expression levels of perilipin, PPAR-α CPT1β, FABP4, and ATGL in adipose tissue of HFD-induced obese mice, as compared to HFD-fed controls, even though it did not register significance. Perillipin has a central role in adipocyte lipolysis and perilipin-mediated activation of HSL increase lipolysis in adipocytes [34]. In the present study, we found that AHR treatment enhanced perilipin and HSL expression compared with control group, thereby contributing to increase of lipolysis in adipose tissues. Therefore, we postulate that AHR-induced lipolysis enhancement might be most dependent on the activation of HSL and LPL, since ATGL is not involved in the effect of AHR on mitochondrial β-oxidation of FFA in adipose tissue. Therefore, PPAR-γ activation by AHR treatment might have promoted HSL and LPL-medicated lipolysis, thereby reducing WAT weight. SREBP-1c is a nuclear transcription factor that regulates the expression of downstream target genes involved in glucose utilization and fatty acid synthesis, such as FAS and SCD-1 [38]. The present study showed that genes expression involved in lipogenesis, SREBP-1c, SCD-1 and FAS were slightly increased by AHR treatment, as compared to control groups. This result might be because AHR potentiated lipolysis rather than lipogenesis in WAT, and alleviated lipogenesis in the liver of HFD-induced mice. Our results were different from those described by Amor et al. [49] who reported that *S*-allylcysteine from aged black garlic decreased adipogenesis through downregulation of gene expression of PPAR-γ, LPL, and HSL in subcutaneous adipose tissue and improved insulin sensitivity in brown adipose tissue. On the other hand, some recent studies have reported that organo-sulfur compounds can ameliorate obesity by enhancing uncoupling protein 1 expression, glucose uptake, oxidative utilization, lipolysis, and fatty oxidation in 3T3-L1 adipocytes [50,51,52,53]. Similarly, the consumption of miso (fermented soybean paste) can enhance lipolysis by increasing two major lipolytic enzymes, HSL and LPL and mRNA expression of PPAR-γ in WAT of HFD induced mice. In addition, hepatic mRNA expression involved in cholesterol synthesis and fatty acid synthesis was decreased by a miso diet [54]. Based on these aforementioned observations, the effects of AHR include suppression of fat accumulation in adipocyte and enhancement of lipolysis in WAT of HFD-induced mice. To explain the lipolysis-promoting activation of AHR, its effect on the PPAR-γ mediated signaling pathway also needs to be investigated in the future.

## 5. Conclusions

AHR treatment reduced lipid accumulation, most probably by inhibiting the expression of adipogenic factors such as C/EBP and LPL in 3T3-L1 adipocytes. In addition, AHR treatment reduced adiposity by preventing increase of WAT mass and ameliorated hyperleptinemia in HFD-induced mice. Lower adiposity might be associated with an induction in expression of lipolytic enzymes such as LPL and HSL medicated by PPAR-γ in WAT. AHR treatment also prevented fatty liver, most probably by inhibiting gene expression involved in lipogenesis such as HMG-CoA reductase, SREBP1/2, FAS, and LDL receptor. Serum TG levels were also lower after AHR treatment which is related to upregulation of genes involved in fatty acid -oxidation, CPT1a. Taken together, this study demonstrates that the anti-obesity effect of AHR was mediated through the downregulation of adipogenesis and lipogenesis as well as potentiating lipolysis in WAT of HFD-induced obese mice.

## Figures and Tables

**Figure 1 nutrients-11-02262-f001:**
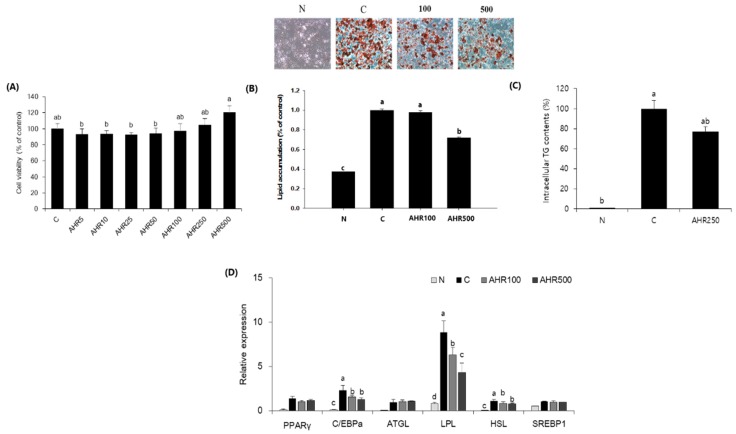
*Allium hookeri* root (AHR) reduces lipid accumulation in 3T3-L1 adipocytes. (**A**) Cell viability of 3T3-L1 preadipocytes treated with AHR for 24 h determined by MTT assay. (**B**) Effect of AHR on lipid accumulation in 3T3-L1 adipocytes determined by Oil Red O staining. (**C**) Intracellular TG contents determined by commercial kit. (**D**) Gene expression after 8 days of incubation of 3T3-L1 adipocytes. Data are expressed as mean ± SD (*n* = 6). * *p* < 0.05, ** *p* < 0.01, and *** *p* < 0.001 vs. control (**C**).

**Figure 2 nutrients-11-02262-f002:**
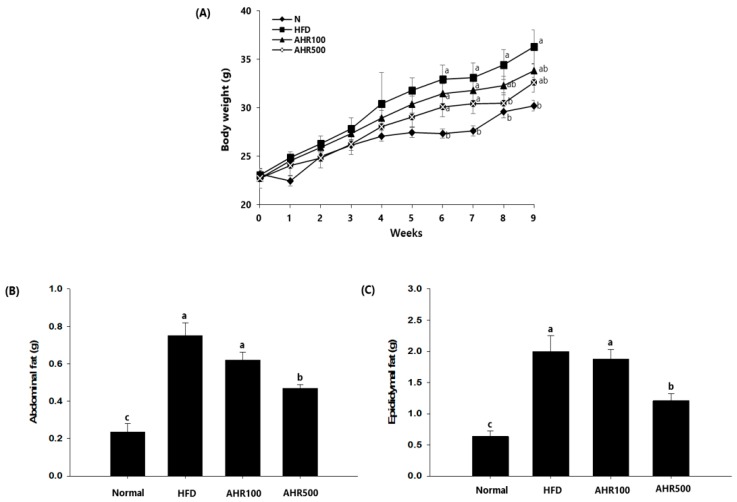
Effects of AHR on body weight changes and fat weight in high-fat diet (HFD)-induced obese mice. Body weight (**A**) was measured during 9 weeks. Abdominal (**B**) and epididymal (**C**) fat weights are shown. Data are expressed as mean ± SEM (*n* = 6–8). Values with different letters are significantly different at *p* < 0.05.

**Figure 3 nutrients-11-02262-f003:**
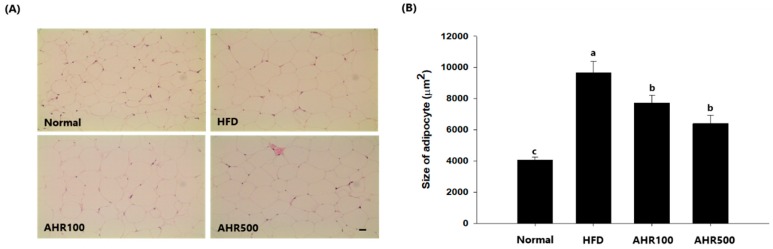
Effects of AHR on fat tissue morphology (**A**) and fat size (**B**) in HFD-induced obese mice. Hematoxylin–eosin staining of epididymal adipose tissue sections from respective mice of each group (scale bar = 100 µm). Original magnification 200×. Data are expressed as mean ± SEM (*n* = 6–8). Values with different letters are significantly different at *p* < 0.05.

**Figure 4 nutrients-11-02262-f004:**
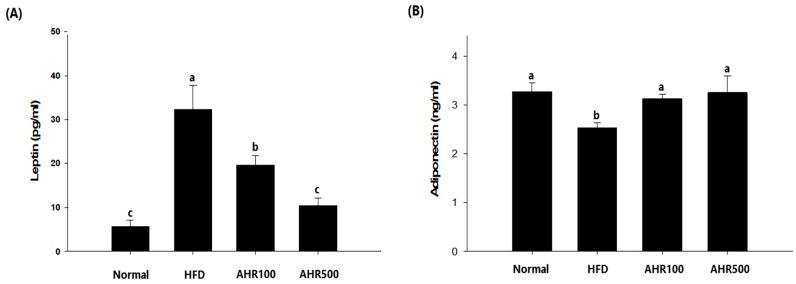
Effects of AHR on serum leptin (**A**) and adiponectin (**B**) concentrations in HFD-induced obese mice. Data are expressed as mean ± SEM (*n* = 6–8). Values with different letters are significantly different at *p* < 0.05.

**Figure 5 nutrients-11-02262-f005:**
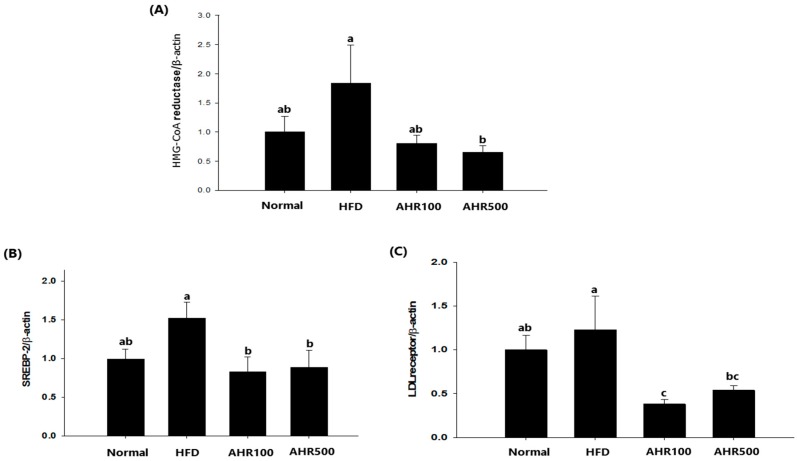
Effects of AHR on hepatic mRNA levels of HMG-CoA reductase (**A**), SREBP-2 (**B**), and LDL receptor (**C**) in HFD-induced obese mice. mRNA expression level was determined by real-time RT-PCR quantification and normalized to that of mRNA *β*-actin. Data are expressed as mean ± SEM (*n* = 6–8). Values with different letters are significantly different at *p* < 0.05.

**Figure 6 nutrients-11-02262-f006:**
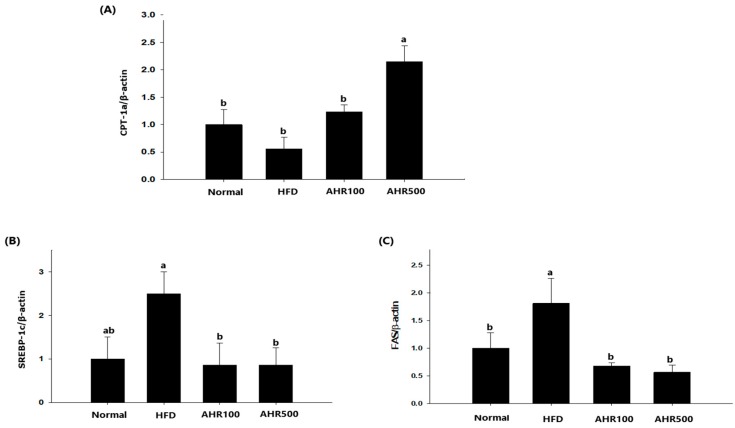
Effects of AHR on hepatic mRNA levels of CPT-1a (**A**), SREBP-1c (**B**), and FAS (**C**) in HFD-induced obese mice. mRNA expression level was determined by real-time RT-PCR quantification and normalized to that of mRNA *β*-actin. Data are expressed as mean ± SEM (*n* = 6–8). Values with different letters are significantly different at *p* < 0.05.

**Table 1 nutrients-11-02262-t001:** PCR primers and conditions used for real-time PCR in 3T3-L1 adipocytes.

Gene	Primer Sequence (5′–3′)
PPAR-γ	5′-GCCCACCAACTTCGGAATC-3′
	5′-TGCGAGTGGTCTTCCATCAC-3′
C/EBPa	5′-GTGTGCACGTCTATGCTAAACCA-3′
	5′-CTCCACTGCCCACCTGTCA-3′
ATGL	5′-AACACCAGCATTCCAGTTCAA-3′
	5′-TTGTGTTGCTTGCCATTCTC-3′
LPL	5′-ACTCGCTCTCAGATGCCCTA-3′
	5′-TTGTGTTGCTTGCCATTCTC-3′
HSL	5′-ACCGAGACAGGCCTCAGTGTG-3′
	5′-GAATCGGCCACCGGTAAAGAG-3′
SREBP-1	5′-ATGATCATCAGCGTAAATGG-3′
	5′-GCCTTTCATAACACATTCCA-3′
β-actin	5′-AGCCTTCCTTCTTGGGTATGG-3′
	5′-CACTTGCGGTGCACGATTGGAG-3′

**Table 2 nutrients-11-02262-t002:** PCR primers and conditions used for real-time PCR in the liver.

Gene	Nucleotide Sequences	Annealing (°C)
HMG-CoA reductase	5′-GTTCTTTCCGTGCTGTGTTCTGGA-3′	60
	5′-CTGATATCTTTAGTGCAGAGTGTGGCAC-3′	
LDL receptor	5′-CTGTGGGCTCCATAGGCTATCT-3′	60
	5′-GCGGTCCAGGGTCATCTTC-3′	
SREBP-2	5′-GTGGAGCAGTCTCAACGTCA-3′	60
	5′-TGGTAGGTCTCACCCAGGAG-3′	
CPT-1a	5′-TCCACCCTGAGGCATCTATT-3′	62
	5′-ATGACCTCCTGGCATTCTCC-3′	
FAS	5′-AGGGGTCGACCTGGTCCTCA-3′	60
	5′-GCCATGCCCAGAGGGTGGTT-3′	
SREBP-1c	5′-CACTTCTGGAGACATCGCAAAC-3′	64
	5′-ATGGTAGACAACAGCCGCATC-3′	
β-actin	5′-AGAGAAGCTGTGCTATGTT-3′	60
	5′-CACAGGATTCCATACCCAAG-3′	

HMG-CoA reductase: 3-hydroxy-3-methylglutaryl CoA reductase, LDL receptor: Low density lipoprotein receptor, SREBP: Sterol regulatory element-binding transcription factor, CPT-1a: Carnitine palmitoyltransferase 1, FAS: Fatty acid synthase.

**Table 3 nutrients-11-02262-t003:** NanoString Target Sequences in white adipose tissue (WAT).

Name	Accession No.	Target Sequence
ABCA1	NM_013454.3	CTCCTTGTCATCTCTAGCCAGGATATTCAGCATCCTCTCCCAGAGCAAAAAGCGACTCCACATAGAAGACTACTCTGTCTCTCAGACAACACTTGACCAA
ATGL	NM_025802.2	ACAGCTCCACCAACATCCACGAGCTTCGCGTCACCAACACCAGCATCCAGTTCAACCTTCGCAATCTCTACCGCCTCTCGAAGGCTCTCTTCCCGCCAGA
Adiponectin	NM_009605.4	GACCACAATGGACTCTATGCAGATAACGTCAACGACTCTACATTTACTGGCTTTCTTCTCTACCATGATACCAACTGACTGCAACTACCCATAGCCCATA
CD36	NM_007643.3	GGGACCATTGGTGATGAAAAAGCAGAAATGTTCAAAACACAAGTGACTGGGAAAATCAAGCTCCTTGGCATGGTAGAGATGGCCTTACTTGGGATTGGAG
CEBP-α	NM_007678.3	AGGAGGACACGGGGACCATTAGCCTTGTGTGTACTGTATGTCGCCAGCCGCTGTTGCTGAAGGAACTTGAAGCACAATCGATCCATCCCAGAGGGACTGG
CPT1-b	NM_009948.2	ACAAGATGTCTCTGGACGCCATCGAACGTGCTGCTTTCTTTGTGACCCTGGATGAAGATTCTCATTGCTACAACCCTGACGATGAGACCAGTCTTAGCCT
DGAT	NM_010046.2	CTATCACTCCAGTGGGTTCCGTGTTTGCTCTGGCATCATACTCCATCATGTTCCTCAAGCTTTATTCCTACCGGGATGTCAACCTGTGGTGCCGCCAGCG
FABP4	NM_024406.2	TCGAAGGTTTACAAAATGTGTGATGCCTTTGTGGGAACCTGGAAGCTTGTCTCCAGTGAAAACTTCGATGATTACATGAAAGAAGTGGGAGTGGGCTTTG
FASN	NM_007988.3	TTCTCCTCTGTAAGCTGCGGGCGTGGTAATGCTGGCCAAACTAACTACGGCTTCGCCAACTCTACCATGGAGCGTATATGTGAACAGCGCAGGCACGATG
GLUT4	NM_009204.2	CTGATGTGTCTGACGCACTAGCTGAGCTGAAGGATGAGAAACGGAAGTTGGAGAGAGAGCGTCCAATGTCCTTGCTCCAGCTCCTGGGCAGCCGCACCCA
HSL	NM_001039507.2	CAGGAGTGCTCTTCTTCGAGGGTGATGAAGGACTCACCGCTGACTTCCTGCAAGAGTATGTCACGCTACACAAAGGCTGCTTCTACGGCCGCTGCCTGGG
LPL	NM_008509.1	CCATGCTGTAACCAAGTCTGGCCTAGAACTAAACTATGTATTTCAGGCTGGCCTTGAACTCTCAACCATCCTGCCTTAGCTTCCTGTGTCCTGGGAGCTT
Leptin	NM_008493.3	CCTATTGATGGGTCTGCCCAAGGCAAACCTAATTTTTGAGTGACTGGAAGGAAGGTTGGGATCTTCCAAACAAGAGTCTATGCAGGTAGCGCTCAAGCTT
PDH	NM_001098231.1	AGTCTGCCACTGTTCTCTGATGCCATGCCAGCACCAACTCAACTGTTTTTTCCTCTCGTCCGTAACTGTGAACTGAGCAGAATCTATGGCACTGCATGTT
PPAR-α	NM_011144.2	GGACTTGAACGACCAAGTCACCTTGCTAAAGTACGGTGTGTATGAAGCCATCTTCACGATGCTGTCCTCCTTGATGAACAAAGACGGGATGCTGATCGCG
PPAR-γ	NM_011146.1	ACCAAGTGACTCTGCTCAAGTATGGTGTCCATGAGATCATCTACACGATGCTGGCCTCCCTGATGAATAAAGATGGAGTCCTCATCTCAGAGGGCCAAGG
Perilipin	NM_001113471.1	TACCAAAGGGAGGGCCATGTCCCTATCCGATGCCCTGAAGGGTGTTACGGATAACGTGGTAGACACTGTGGTACACTATGTGCCGCTTCCCAGGCTGTCC
SCD1	NM_009127.3	GTGTTGCCTGGGTTGCCAGTTTCTTTCGTGGCTGGGCAGGAACTAGTGAGGTTGAGGGGCAGTGTCTGTAAGTAGCTGCTAAGAGGTGCATTTCCAGATG
SREBP-1c	NM_011480.1	GACTACATCCGCTTCTTGCAGCACAGCAACCAGAAGCTCAAGCAGGAGAACCTGACCCTACGAAGTGCACACAAAAGCAAATCACTGAAGGACCTGGTGT
β-actin	NM_007393.1	CAGGTCATCACTATTGGCAACGAGCGGTTCCGATGCCCTGAGGCTCTTTTCCAGCCTTCCTTCTTGGGTATGGAATCCTGTGGCATCCATGAAACTACAT

ABCA1: ATP-binding cassette transporter, ATGL: Adipose triglyceride lipase, CD36: cluster of differentiation36, CEBP-α: CCAAT/enhancer-binding protein alpha, CPT1-b: Carnitine palmitoyl transferase 1-b, DGAT: Diglyceride acyltransferase, FABP4: fatty acid binding protein 4, FASN: Fatty acid synthase, GLUT4: Glucose transporter type 4, HSL: Hormone-sensitive lipase, LPL: Lipoprotein lipase, PDH: Pyruvate dehydrogenase, PPAR-α: Peroxisome proliferator-activated receptor alpha, PPAR-γ: Peroxisome proliferator-activated receptor gamma, SCD1: Stearoyl-CoA desaturase-1, SREBP-1c: Sterol regulatory element-binding protein 1c.

**Table 4 nutrients-11-02262-t004:** Effect of AHR on body weight, food intake, and feeding efficiency ratio (FER) in HFD-induced mice.

Group	Body Weight Gain (g)	Food Intake (g)	FER *
Normal	7.06 ± 0.56 ^b^	2.76 ± 0.35 ^a^	2.66 ± 0.33 ^c^
HFD	12.99 ± 3.98 ^a^	2.32 ± 0.16 ^b^	5.71 ± 1.79 ^a^
AHR100	11.31 ± 1.58 ^a^	2.33 ± 0.17 ^b^	4.96 ± 0.76 ^ab^
AHR500	10.20 ± 0.87 ^a^	2.41 ± 0.15 ^b^	4.23 ± 0.43 ^b^

Data are expressed as mean ± SEM (*n* = 6–8). Values with different letters are significantly different at *p* < 0.05. * FER, Feeding efficiency ratio = Body weight gain (g)/food intake (g) × 100.

**Table 5 nutrients-11-02262-t005:** Effect of AHR on tissue weight in HFD-induced mice.

Group	Liver (g)	Heart (g)	Kidney (g)
Normal	0.97 ± 0.02 ^a^	0.13 ± 0.01	0.33 ± 0.01
HFD	0.99 ± 0.04 ^a^	0.12 ± 0.00	0.33 ± 0.01
AHR100	0.91 ± 0.02 ^ab^	0.12 ± 0.00	0.32 ± 0.01
AHR500	0.82 ± 0.04 ^b^	0.13 ± 0.01	0.034 ± 0.01

Data are expressed as mean ± SEM (*n* = 6–8). Values with different letters are significantly different at *p* < 0.05.

**Table 6 nutrients-11-02262-t006:** Effect of AHR on serum lipid concentration in HFD-induced obese mice.

Group	T-Chol (mg/dL)	TG (mg/dL)
Normal	95.78 ± 4.22 ^a^	97.64 ± 7.33 ^b^
HFD	117.64 ± 8.02 ^a^	116.80 ± 9.87 ^a^
AHR100	119.21 ± 1.99 ^a^	117.98 ± 2.84 ^a^
AHR500	109.61 ± 4.62 ^a^	105.52 ± 2.52 ^ab^

Data are expressed as mean ± SEM (*n* = 6–8). Values with different letters are significantly different at *p* < 0.05.

**Table 7 nutrients-11-02262-t007:** Effect of AHR on lipid metabolism related gene levels in epididymal adipose tissue of HFD-induced obese mice.

Gene Function	Name	Fold Change			
		AHR100 * vs. HFD	*p*-Value	AHR500 ** vs. HFD	*p*-Value
Cholesterol efflux	ABCA1	−1.13	0.152	−1.00	0.978
Regulates the metabolism of lipid and glucose	Adiponectin	−1.02	0.870	1.21	0.217
Imports fatty acids	CD36	1.08	0.664	1.14	0.468
Induction of adipogenesis	CEBP-α	−1.10	0.515	1.23	0.178
Adipocyte differentiation	PPAR-γ	1.19	0.415	1.79	0.023
Fatty acid β-oxidation	CPT1-β	−1.06	0.666	1.02	0.914
	FABP4	−1.08	0.500	1.13	0.312
	PPAR-α	−1.02	0.912	1.55	0.073
Glucose transporter	GLUT4	−1.01	0.934	1.31	0.161
TG lipolysis	HSL	1.03	0.894	1.62	0.031
	LPL	1.16	0.291	1.44	0.028
	ATGL	1.08	0.813	1.78	0.094
Regulate appetite	Leptin	1.31	0.259	−1.38	0.176
Adipocyte lipolysis	Perilipin	1.05	0.814	1.53	0.056
Lipogenesis	SCD1	1.04	0.873	1.70	0.067
	SREBP-1c	1.02	0.911	1.18	0.325
	PDH	−1.05	0.774	1.27	0.085
	DGAT	1.05	0.774	1.61	0.045
	FASN	1.44	0.084	1.54	0.145
House keeping	β-actin	1		1	

ABCA1: ATP-binding cassette subfamily A member 1, ATGL: Adipose triglyceride lipase, CD36: Cluster of differentiation 36, CEBP-α; CCAT/enhancer-binding protein, CPT1-β: Carnitine palmitoyltransferase 1, DGAT: Diacylglycerol acyltransferases, FABP4: Fatty acid-binding protein, FASN: Fatty acid synthase, GLUT4: Glucose transporter type 4, HSL: Hormone-sensitive lipase, LPL: Lipoprotein lipase, PDH: Pyruvate dehydrogenase, PPAR-α: Peroxisome proliferator-activated receptor, SCD1: Stearoyl-CoA desaturase-1, SREBP-1: Sterol regulatory element-binding transcription factor. * AHR100: HFD + 100mg/kg B.W. water extract of *Allium hookeri* root. ** AHR500: HFD + 500mg/kg B.W. water extract of *Allium hookeri* root.

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
