# Peer review of "Allium hookeri Root Extract Inhibits Adipogenesis by Promoting Lipolysis in High Fat Diet-Induced Obese Mice"

_nutrients, 2019, doi:10.3390/nu11102262_

Round 1

Reviewer 1 Report

The authors investigated the anti-adipogenic and anti-obesity effects of allium hookeri root water extract (AHR). AHR inhibited lipid accumulation in adipocytes, and the expression of LPL and C/EBPa was reduced by AHR treatment. Moreover, in vivo study, AHR decreased body weight gain, adipose tissue weight, and adipocyte cell size in HFD-fed mice. In addition, AHR lowered the expression of HMG-CoA reductase, SREBP-2, LDL-R, and FAS genes in liver. Thus, they concluded that AHR improved obesity by modifying the expression levels of the genes involved in adipogenesis, lipogenesis, and lipolysis in WAT and liver of obesity. The study is incomplete. The organization of manuscript should be improved. There is no mechanistic data. Description of method part is not enough. There are many concerns that should be improved. Moreover, English should be improved. There are typos and grammatical errors.

Major

The method for cell viability assay should be shown. The primer sequences for b-actin. used in real-time PCR should be shown. Is it really real-time PCR? Amplification condition is as if the standard PCR. If it is real-time PCR, enzyme for PCR, probably used SYBR Green should be indicated. Line 109; “Adipocytes differentiated for 4 or 7 days were treated with AHR” means that differentiated adipocytes were treated with AHR? If so, how long were the differentiated cells treated with AHR? Please indicate. Why was the MTT assay performed for 24 hours? Adipocyte differentiation in the presence of AHR was for 8 days. The toxicity of AHR toward 3T3-L1 cells should be examined for 8 days. The conclusion from Fig. 1 is difficult to understand. The reduction of lipid accumulation is derived from a decrease in adipogenesis or an increase in lipolysis, or both? Moreover, the expression levels of the other adipogenic, lipogenic, and lipolytic genes should be investigated. Only two genes (C/EBPa and LPL) are too little. Have you checked the liver toxicity of AHR? The order of figures is strange. Fig. 6 should be just after Fig. 2. Why did the authors investigate livers in HFD-fed AHR-administered mice? The purpose should be clearly explained. Moreover, only expression data were shown in liver. The issue should be clearly described. There are no mechanistic data. The change of expression levels of many genes were shown. After all, what is the possible target of AHR? If expression of PPARg was changed the mRNA levels of a variety of genes involved in the adipogenic, lipogenic, and lipolytic genes were altered. AHR is a ligand for PPARg? Possible mechanism should be clearly explained in Discussion part. English should be improved. There are typos and grammatical errors. Please carefully check the manuscript prior to submission.

Minor

“bovine calf serum (BCS)”; line 80 is the same as “calf serum”; line 86? Line 92; “Cells were treated with 0, 100, or 500 μg/mL of AHR extract every day.” means that AHR was added every day? Description is ambiguous. Please improve the description.

Author Response

Q1: The method for cell viability assay should be shown.

A1: We have revised this manuscript with the method for cell viability assay.

Q2: The primer sequences for b-actin. used in real-time PCR should be shown. Is it really real-time PCR? Amplification condition is as if the standard PCR. If it is real-time PCR, enzyme for PCR, probably used SYBR Green should be indicated.

A2: We have revised this manuscript according to your suggestions.

Q3: Line 109; “Adipocytes differentiated for 4 or 7 days were treated with AHR” means that differentiated adipocytes were treated with AHR? If so, how long were the differentiated cells treated with AHR? Please indicate.

A3: We have revised this manuscript with more information.

Q4: Why was the MTT assay performed for 24 hours? Adipocyte differentiation in the presence of AHR was for 8 days.

A4: We have revised this manuscript with more information.

Q5: The toxicity of AHR toward 3T3-L1 cells should be examined for 8 days. The conclusion from Fig. 1 is difficult to understand. The reduction of lipid accumulation is derived from a decrease in adipogenesis or an increase in lipolysis, or both? Moreover, the expression levels of the other adipogenic, lipogenic, and lipolytic genes should be investigated. Only two genes (C/EBPa and LPL) are too little. Have you checked the liver toxicity of AHR?

A5: We have revised this manuscript with more information and cited reference showing hepatoprotective activities of AHR against oxidative stress.

Q6: order of figures is strange. Fig. 6 should be just after Fig. 2.

A6: We have revised this manuscript.

Q7: Why did the authors investigate livers in HFD-fed AHR-administered mice? The purpose should be clearly explained. Moreover, only expression data were shown in liver. The issue should be clearly described.

A7: We have revised this manuscript.

Q8: There are no mechanistic data. The change of expression levels of many genes were shown.

A8: We have revised this manuscript with more information.

Q9: After all, what is the possible target of AHR? If expression of PPARg was changed the mRNA levels of a variety of genes involved in the adipogenic, lipogenic, and lipolytic genes were altered. AHR is a ligand for PPARg? Possible mechanism should be clearly explained in Discussion part.

A9: We have revised this manuscript.

Q10: English should be improved. There are typos and grammatical errors. Please carefully check the manuscript prior to submission.

A10: We have revised this manuscript by double checking typos and grammatical errors.

Q11: “bovine calf serum (BCS)”; line 80 is the same as “calf serum”; line 86?

A11: We have revised this manuscript.

Q12: Line 92; “Cells were treated with 0, 100, or 500 μg/mL of AHR extract every day.” means that AHR was added every day? Description is ambiguous. Please improve the description.

A12: We have revised this manuscript by improving description.

Thank you so much valuable comments to improve this manuscript.

Reviewer 2 Report

Comments to the atuhors: In this manuscript, there are serious weak points and you should revise in the points picked out below.   1. The author should more explain about your sample, Allium hookeri root in the introduction section.
2. The author have to describe the statistical method used in this study.

3. In this study, mice were given free access to the experimental diet containing AHR. The author should show how the dose of AHR was adjusted at mg per kg body weight.
4. The author should give more informaiton about perilipin and SCD1 relatee to adipocyte lipolysis (shown on line 257) with cited references and discuss about those relation in your results.  

Author Response

 Q1. The author should more explain about your sample, Allium hookeri root in the introduction section.

A1: We have revised this manuscript with more explaining about our sample, Allium hookeri root in the introduction section.

Q2. The author have to describe the statistical method used in this study.

A2: We have revised this manuscript by describing the statistical method.

Q3. In this study, mice were given free access to the experimental diet containing AHR. The author should show how the dose of AHR was adjusted at mg per kg body weight.

A3: We have shown the information about the dose of AHR in this manuscript.   

Q4. The author should give more informaiton about perilipin and SCD1 relatee to adipocyte lipolysis (shown on line 257) with cited references and discuss about those relation in your results.  

A4: We have revised this manuscript with adding references and discussing about those relation in our results.

Reviewer 3 Report

very good job. Congratulation

Author Response

Thank you for the kind comments and suggestions for us. 

Round 2

Reviewer 1 Report

Thank you for the response to my concerns. The revised manuscript was improved. However, there are some additional minor concerns that should be addressed. Please see below.

What is the difference of the primers listed in 2.4, and Table 1? In the first place, what is the difference between 2.4. and 2.8.? Moreover, the primers in 2.4. should be shown as Table. Line 394: “HSL, a major enzyme for hydrolysis of TG, is up-regulated by PPAR-γ or its agonists. “ should be improved. Because PPARg is a ligand-dependent transcription factor. Thus, “PPAR-γ or its agonists” is so strange. “or its agonists” should be deleted.

Author Response

The authors wish to thank the astute reviewers for their support and valuable advice for revision of the paper. We were happy to address all issues raised as follows:

Q: What is the difference of the primers listed in 2.4 and Table1? In the first place, what is the difference between 2.4 and 2.8? Moreover, the primers in 2.4. should be shown as Table.

A: The primers listed in 2.4. presents genes involved in adipogenesis in 3T3-L1 adipocytes and 2.8. presents genes involved in adipogenesis, lipogenesis, and lipolysis in white adipose tissue of HFD-induced obese mice. As your comments, the primers in 2.4 revised to Table 1.

Q: Line394. “HSL, a major enzyme for hydrolysis of TG, is up-regulated by PPAR-γ or its agonists.” Should be improved. Because PPAR-γ is a ligand-dependent transcription factor. Thus, “PPAR-γ or its agonists” is so strange. “or its agonists” should be deleted.  

A: We revised it as your detailed comment.

Reviewer 2 Report

Thank you for the responses to my concerns.
The revised manuscript was improved, however there are some minor concerns that should be addressed.
Please you response the points shown below.

1) I think that the readers are confused about the primers' information for real-time PCR.
 Please authors clearly explain why different primers are used for 3t3-l1 cells and mice studies, and shows that information in Table 1 is for animal experiment.

2) I recommend that the authors statistically analyze the data related with 3T3-L1 adipocyte by multiple range-test instead of t-test.
  3) In line 406, the author revised "AHR elevated mRNA expression ...., SCD1", but also showed "it did not significantly increase...SCD1...." in line 410. I confused those explanations.   4) The athors included the SCD1 and SREBP-1c genes as related with adipocyte lipolysis (especially SCD1). Please give its related information more in the discussion section.

Author Response

The authors wish to thank the astute reviewers for their support and valuable advice for revision of the paper. We were happy to address all issues raised as follows:

Q: I think that the readers are confused about the primers’ information for real-time PCR. Please authors clearly explain why different primers are used for 3T3-L1 cells and mice studies, and shows that information in Table 1 is for animal experiment.

A: I totally agree with your concern. 3T3-L1 study was conducted by Jang JY as a preliminary study with some limited primers at first and then mice study was conducted by Lee MJ, respectively. The primers used in mice study have wide ranges of genes involved in adipogenesis, lipogenesis and lipolysis, which includes genes used in cell study.

Q: I recommend that authors statistically analyze the data related with 3T3-L1 adipocyte by multiple range-test instead of t-test.

A: Statistical analysis for the 3T3-L1 adipocyte study was used ANOVA (SPSS 12.0 software). It was revised in Statistical analysis part and Fig 1.

Q: In line 406, the author revised “AHR elevated mRNA expression…., SCD1”, but also showed “it did not significantly increase…SCD1…” in line 410. I confused those explanations.

A: We have made it more clearly.

Q: The author included the SCD1 and SREBP1c genes as related with adipocyte lipolysis (especially SCD1). Please give its related information more in the discussion section.

A: In line 420, we have revised it with reference.